# Recent Advances in Raman Spectral Imaging in Cell Diagnosis and Gene Expression Prediction

**DOI:** 10.3390/genes13112127

**Published:** 2022-11-16

**Authors:** Tomonobu M. Watanabe, Kensuke Sasaki, Hideaki Fujita

**Affiliations:** 1Department of Stem Cell Biology, Research Institute for Radiation Biology and Medicine, Hiroshima University, 1-2-3 Minami-ku, Hiroshima 734-8553, Japan; 2Laboratory for Comprehensive Bioimaging, RIKEN Center for Biosystems Dynamics Research (BDR), 2-2-3 Minatojima-minamimachi, Kobe 650-0047, Japan

**Keywords:** spectroscopy, machine learning, non-linear optics

## Abstract

Normal and tumor regions within cancer tissue can be distinguished using various methods, such as histological analysis, tumor marker testing, X-ray imaging, or magnetic resonance imaging. Recently, new discrimination methods utilizing the Raman spectra of tissues have been developed and put into practical use. Because Raman spectral microscopy is a non-destructive and non-labeling method, it is potentially compatible for use in the operating room. In this review, we focus on the basics of Raman spectroscopy and Raman imaging in live cells and cell type discrimination, as these form the bases for current Raman scattering-based cancer diagnosis. We also review recent attempts to estimate the gene expression profile from the Raman spectrum of living cells using simple machine learning. Considering recent advances in machine learning techniques, we speculate that cancer type discrimination using Raman spectroscopy will be possible in the near future.

## 1. Raman Spectroscopy of Mammalian Cells

Biological substances are composed of various biomolecules, each of which has a unique molecular structure with various vibrational modes. Light irradiated on these substances is scattered after the energy is transferred; corresponding to these vibrational modes between the substances and the light is the Raman scattering phenomenon. The Raman scattered light emitted by Raman active substances is the light that has been modulated by all the vibration modes of the constituent molecules. Hence, the obtained spectrum of light contains information on all the molecular constituents present and their physical state, making it possible to determine the molecular composition of the subjects. However, in the case of biological samples, such as cells, is more complex. The Raman spectrum emitted by a cell comprises the vibrational modes of a wide variety of molecular species, including proteins [1,2,3,4], lipids [5], and nucleotides [6,7], which are superimposed in too complex a manner to be resolved [8,9]. However, Raman scattering spectra can be acquired non-invasively and can be used as a quantifiable parameter to describe the function and state of cells [10]. To achieve this, the technological development of signal- and information-processing technologies [11], including chemometrics and machine learning [12], is indispensable. Machine learning methods turn the complexity of Raman scattering spectra of cells from a weakness to an advantage [8].

## 2. Raman Observation of Living Cells

Raman scattered light is weak relative to irradiated light. Therefore, in practice, strong light irradiation is required to obtain a good signal; however, this causes photodamage to the cell. When cells are exposed to strong light, reactive oxygen species are internally produced, which can destroy DNA, mitochondria, and other organelles, thereby leading to cell death [13]. Additionally, metabolic activity may change as a result of resistance to light irradiation, which is included as false signals in the acquired Raman scattering spectrum. To correctly measure the Raman spectra of cells, it is essential to collect weak signals while keeping the cells alive by minimizing light-induced damage. Since the 1980s, with the increase in sensitivity of multichannel spectrometers and detectors and the significant improvement of filter technology, it is now possible to analyze spectra even with weak Raman scattering light [14]. Raman scattering spectroscopy of cells is currently widely used in the fields of cell biology and medicine [15,16].

Katsumasa Fujita, and colleagues constructed line-scanning Raman spectroscopic microscope specialized for observing living cells [17,18]. This line scanning configuration dramatically increased the scanning speed to ~100× *g* compared to point scanning confocal Raman microscope [19]. Because cells contain numerous molecules that emit autofluorescence, the detected spectrum also includes the autofluorescence of the cells. If the cells are attached to a substrate, such as a glass slide, the signal derived from the glass will also be included. Additionally, microstructures that cause scattering in the cell and multiple scattering that occurs between the cell and the substrate are also included in the detected spectrum. Methods to remove these background signals are a subject of research and development [20]. The objective is to estimate and remove the background signal via polynomial approximation and extract a clear narrow peak or extract only the cell-derived signal by subtracting the signal of the substrate.

Raman scattering in the mouse fibroblast cell line NIH3T3, show a complex spectrum (Figure 1A) with some peaks where the resulting vibrational mode is clear. For example, the peak at 1340 cm^−1^ was mainly attributed to the CH_2_ stretching mode in the main lipid chain (Figure 1B, blue). The peaks at 753 cm^−1^, 1131 cm^−1^, and 1585 cm^−1^ are attributed to the protein cytochrome c (Figure 1B, red), and the peak at 1686 cm^−1^ is attributed to peptide bonds in various proteins (Figure 1B, green). By assigning the signal intensities of three of these characteristic peak values (753 cm^−1^, 1686 cm^−1^, and 1340 cm^−1^) to the three blue-red-green colors, a pseudo-color image reflecting the Raman scattering spectrum of the cell line can be created (Figure 1B). Different cell types and conditions contain different ratios of constituent molecules. Therefore, the Raman scattering spectra should exhibit different shapes depending on the cell type. Raman spectra from three cell lines of mouse origin: fibroblasts (NIH3T3), mammary epithelial cells (EpH4), and hepatoma cells (Hepa1-6) were collected [9]. By comparing the pseudo-color images created based on the Raman scattering spectra emitted from each cell, it was confirmed that the color tones were different, that is, the Raman scattering spectra were different (Figure 1C). In addition to differences in cell line types, the changes in the color of the pseudo-color images were observed before and after cell differentiation, both in the case of neuronal differentiation (mouse neuroblasts (Neuro 2a)) (Figure 1D) and adipocyte differentiation (mouse fibroblasts (3T3-L1)) (Figure 1E). This confirms that Raman scattering spectra reflect the cell type and state [9]. Other groups successfully discriminated cell lineages in co-culture of MDCK and CHO cells using partial least-squares discriminant analysis, showing the applicability of Raman spectroscopy in non-label cell type discrimination [21].

The signal intensity of selective peaks in Raman scattering spectra can be used to evaluate the function and state of cells, although the application is limited because of the large photodamage. For example, cytochrome c, a heme protein bound to the inner mitochondrial membrane, is one of the few molecules that show multiple clear peaks in the complex Raman scattering spectrum. Cytochrome c shows different structures between oxidized and reduced forms, but the Raman scattering intensity of reduced cytochrome c is higher than that of oxidized cytochrome c. Based on this property, malarial infection of red blood cells can be detected using the Raman scattering spectrum [22]. Additionally, when cells undergo apoptosis, cytochrome c is released from the mitochondria into the cytoplasm, which can be observed using Raman scattering spectroscopy [18]. Because Raman spectroscopy signals are far weaker than fluorescence signals, their temporal resolution is much lower than that of fluorescence observation. However, the risk of artifacts due to fluorescent dye staining or other effects was eliminated. By irradiating cells with light and analyzing the scattered light, cell types and conditions can be identified and separated using Raman scattering spectroscopy without the need for staining or manipulation [23].

## 3. Use of Machine Learning on Raman Spectrum Analysis

In Raman scattering spectra emitted by cells, both sharp peaks and peaks that are partially overlapped or broad to be identified are often observed. For example, although collagen and glycogen do not show significant specific peaks, they do show characteristic spectral shapes, which can be analyzed using principal component analysis and reverse convolution. Cells produce collagen and glycogen as resistance responses to irradiation, and this feature can be used to evaluate the radioactivity tolerance of cells using Raman scattering spectra [24]. Discrimination of cell type/function/state using principal component analysis and reverse convolution has been widely applied, starting from the imaging diagnosis of cancer cells in the past [25] to the identification of peripheral nerve tissue, including myelinated and unmyelinated nerves, to avoid nerve damage after surgery [26]. There are already examples of its use in medical practice, such as image-guided surgery, to specifically identify cancer cells during cancer tissue removal surgery [27].

In the pseudo-color image obtained using the peak values of the Raman scattering spectra, it was clear that there were differences among the three cell lines (Figure 1C). However, the averaged Raman spectrum obtained from the cell only shows a seemingly minor difference (Figure 2A). This difference is due to the fact that in the pseudo-color image, the contrast ratio was adjusted to emphasize the difference. However, differences were also observed in the small and broad peaks other than the characteristic peaks used in the pseudo-color image. In decomposition analyses, such as pseudo-color imaging and detection of characteristic peaks, only a part of the information in the Raman spectrum is used. Thus, the comprehensiveness of the Raman scattering spectrum cannot be effectively exploited. Therefore, multivariate analysis such as principal component analysis is often used to measure Raman scattering spectra of cells, which can be analyzed while retaining as much information as possible. Principal component analysis is among the simplest machine learning methods available, wherein data described by a large number of variables are reduced to a small number of independent variables for analysis. When the Raman scattering spectra obtained from the three cell lines were projected as points on a three-dimensional coordinate axis using principal component analysis, each spectrum formed a cluster with some distribution (Figure 2B). This result shows that the differences in spectra depending on the cell type and the heterogeneity of cell states can be visualized using principal component analysis. Thus, by using principal component analysis, differences can be evaluated in more detail by performing an analysis that captures the entire picture of the spectral shape, rather than selectively analyzing the signal intensity of specific peaks.

Visualization of cellular heterogeneity can aid in the physical understanding of live phenomena. Attempt were made to visualize the process of state transitions during cell differentiation using Raman scattering spectra [28]. Raman spectra of C2C12 mouse myoblasts (a model cell line for studying muscle differentiation) over time were measured and a principal component analysis during cell differentiation were performed. Pseudo-color images of the Raman spectral peaks showed that the Raman spectra of the cells changed over time before differentiation (myoblast), immediately after induction of differentiation (confluent), 3 days after induction of differentiation (differentiation), and upon completion of differentiation into myotubes (myotubes) (Figure 2C). This three-dimensional distribution represents the existence probability distribution of the cell states. The potential energy that represents the cell state can be visualized by plotting the logarithm of the reciprocal of the probability distribution. C2C12 cells immediately after differentiation were narrowly distributed in two-dimensional space (Figure 2D, left). Three days after induction of differentiation, when C2C12 cells showed an elongated morphology, their distribution on the same coordinate widened (Figure 2D, middle), that is, the potential became shallower and wider. After completion of differentiation, the state of the cells fell into a trough that differed from the trough of the potential before differentiation (Figure 2D, right). These observations indicate that C2C12 cells have a stable cell state before and after the completion of differentiation, whereas their internal state is fluctuating during the transition process of cell differentiation. Similar results have been observed in mouse embryonic stem cells [2], and which indicates that these results are not unique to C2C12 cells. Cell differentiation monitoring by Raman spectroscopy was reported in various models such as cardiomyocyte differentiation [29], neuronal differentiation [30], or differentiation of organoids [31].

Discriminant analysis is one of the simplest supervised machine-learning methods [32]. Discriminant analysis is a method that sets up criteria and axes to separate groups, and calculates the similarity to each group as a score for data that has not been classified. T cells, which are at the center of the biological immune response, are activated by recognizing specific antigen-presenting cells. Stimulation with anti-CD3 and anti-CD28 antibodies mimics the activation of T cells in vitro. First, Raman scattering spectra are collected on T cells before stimulation and 48 h after stimulation; we assume that T cells were fully activated (Figure 2E), and these were used as teacher data for discriminant analysis. From the Raman scattering spectra after an arbitrary elapsed time following stimulation, a score was calculated based on the teacher data (Figure 2F). The scores calculated by measuring the Raman scattering spectra of T cells after 2, 6, 12, 24, and 48 h increased with increasing time after stimulation (Figure 2G). Immune cell activation was assessed by flow cytometry. Upon activation, immune cells expressed specific proteins on the cell surface, which resembled the activation time course obtained by Raman scattering spectra (Figure 2H). Using the discriminant analysis method, the transition process of the cell state can be evaluated without staining using the Raman scattering spectrum. In this manner, the complex Raman scattering spectra of a cell can be effectively used as a “cell fingerprint” using machine learning in combination. Because of the high-content data obtained by Raman spectroscopy, it is compatible to other machine learning methods such as random-forest classification, support vector machine or k-means classification and being used in cancer diagnosis or evaluation of cancer treatment [33,34,35]. Use of Raman spectra in cancer discrimination is recently becoming popular in combination with machine-learning, including artificial intelligence [35,36,37], which is expected to further develop in the near future.

## 4. Utilization of Image Obtained by Raman Spectral Peaks

Morphology of cultured cells are often different between cell types, and discrimination of cell types or state can be possible using bright field or fluorescent image of cells with the use of machine learning [38,39], and more recently by aid of deep learning [40,41]. In Raman spectroscopy of cells, it is necessary to efficiently collect weak Raman scattered light emitted by the cells to minimize light-induced damage [42,43]. Therefore, most examples of Raman scattering spectroscopy of cells are measurements conducted using microscopes equipped with high numerical aperture objectives. Confocal optics is often applied in more practical cases [44,45,46]. In confocal optics, laser light is focused on the cell, Raman light is generated from the molecules in the focused spot, and the light is focused again on the incident pinhole of the spectrometer to be detected as a spectrum. By scanning the position of the focusing spot in the focal plane, a hyperspectral image can be acquired wherein each pixel contains spectral information. Thus, it is possible to construct a pseudo-color image. Therefore, using information from hyper “Raman” spectral images, we compared the accuracy of discrimination of cell types for Raman spectra only, images composed of specific peaks alone, and both [47].

Raman spectra from three mouse-derived hepatocellular carcinoma cell lines (Hepa1-6), neuroblastoma (Neuro2A), and mesenchymal stem cells (MSC) were collected, which clearly show differences in Raman scattering spectra and have distinctive shapes [47]. The differences between the three cell lines were clearly visible when the signal intensities of the cytochrome c-derived peak (753 cm^−1^), protein-derived peak (1687 cm^−1^), and lipid-derived peak (1340 cm^−1^) were used to create a pseudo-color image (Figure 3A). However, when the entire Raman spectrum averaged over the cell region was compared, it was very similar, indicating that there were only a few spectral peaks that contributed to the discrimination of the three cell lines (Figure 3B). Three combinations were compared in this study: (1) image information alone, (2) Raman spectrum information alone, or (3) both, and set up a spatial axis where the difference between the three cell lines could be best expressed by principal component analysis, and investigated the discrimination accuracy by performing discriminant analysis on that principal component space. The accuracy of discriminant analysis of the three cell lines using the average spectrum alone was only 83% (Figure 3C). For analysis using image information, image features of the 300 spatial patterns were obtained using the 11 mathematical algorithms described by Orlov et al. [48]. When discriminant analysis was performed using these features, the accuracy was as high as 95% (Figure 3C). This is a natural result as three species with clear differences in shape were used in the experiment; however, 5% of the patterns remained misidentified. By using both, that is, all the information in the hyperspectral images, the discrimination accuracy reached 100% (Figure 3C). This indicates that the spectral information and spatial distribution information contribute to discrimination in both common and complementary ways.

Thus, it was confirmed that the discrimination accuracy was indeed improved by using both spectral and spatial distribution information; however, the amount of computation increased dramatically. Therefore, the next step was to investigate the effective use of the image features for discrimination. Using a color image created with three spectral peaks, discriminant analysis were performed using only the ten image features with the highest discrimination contribution out of 300 features and found that the discrimination accuracy was 85%; however, using 50 features, the discrimination accuracy reached 97%. Even for images consisting of a single specific peak, the discrimination accuracy reached 90% using the peak derived from cytochrome c (753 cm^−1^); however, the information from the other peaks further improved the discrimination accuracy. Thus, by appropriately selecting the spectral peaks and image features to be used, the number of calculations can be effectively reduced. Recently, by using stimulated Raman scattering (SRS) microscopy which enables fast acquisition of specific Raman peaks, SRS-image based sorting was reported opening the door to non-label cell sorting [49].

## 5. Relationship between Raman Spectra and Gene Expression

The results of the abovementioned study imply that the spectral shape of the Raman spectrum from the cell is patterned according to the state and function of the cell, although the spectral shape becomes so complex that it is difficult to decompose and analyze. Because Raman scattered light comprises all the molecular vibration modes of the observed object, the spectral shape is determined by the type and content ratio of the molecular vibration modes of the observed object. The number of molecular vibration modes in a cell is large but not infinite. This is because the types of molecules that constitute a cell are determined by the types of compounds inside the cell, such as nucleic acids, proteins, and lipids, and are finite. The content ratio of intracellular compounds depends on the type and function of cells. As the type and function of a cell are defined by its gene expression pattern, it can be expected that the Raman spectrum of a cell is correlated with its gene expression pattern. A cell is an assembly of a vast number of elements of various types, such as proteins, genes, and metabolic molecules, interacting in a complex manner; however, the number of functions and states that a cell can adopt is limited. For example, a mammalian cell has more than 20,000 genes, but because cell functions are a result of the complex interaction of these genes, the cell functions can be aggregated and classified into only ~300 different types [50]. Both Raman spectra and gene expression patterns are complex; however, they are interrelated through a finite number of cellular functions.

Although the above hypothesis makes sense, it has not been proven based on actual mammalian cells. However, attempts were made in simpler organism with less genes; *Escherichia coli* (*E. coli*) which only have 4400 genes, to confirm the feasibility of the above assumption [16]. Unlike mammalian cells, the Raman scattering spectra of *E. coli* can be obtained while the cells float in the culture medium. Raman spectra from floating cells are less likely to be contaminated by signals from the glass substrate, allowing for a more quantitative analysis. Gene expression patterns of 11 drug-resistant *E. coli* strains were collected and created paired data corresponding to the Raman spectra acquired simultaneously (Figure 4A). Using these data pairs, correlation spectra were created by calculating the correlation coefficient between each spectral peak and the gene expression. Correlation between the increase or decrease of a certain peak value in the Raman scattering spectrum and the increase or decrease of a certain gene’s expression were reported [51]. For example, the expression of *cyoA* and *cyoD*, genes related to protein synthesis, was negatively correlated with the peak intensities at 730 cm^−1^ and 1476 cm^−1^ (Figure 4B). Alternatively, the expression of genes related to folate synthesis, *floA* and *floC*, was positively correlated with the peak intensity at 752 cm^−1^ and negatively correlated with the peak intensity at 1450 cm^−1^ (Figure 4C). The peak at 1450 cm^−1^ is associated with CH2 deformation. Because there is a correlation between each spectral peak and each gene expression, it is possible, by regression, to estimate the gene expression profile from the acquired Raman spectra. Here, we estimated the gene expression profiles of unknown *E. coli* strains using the partial least squares regression method and found that gene expression could be estimated with an accuracy of 85–90% for all cell lines. However, Raman scattering spectrum analysis is not a suitable substitute for gene expression analysis. This was because the two were mapped through patterning. For example, it is not possible to predict gene expression that is not associated with patterning. There is another important implication from the results of this experiment. The Raman scattering spectrum of a cell depends strongly on its metabolic network. Using this method, it is possible to analyze metabolism and gene expression as a cross-network. In the near future, we expect that Raman scattering spectroscopy of cells will be used in combination with omics technologies.

## 6. Conclusions

Here, we discuss the applicability of Raman spectroscopy in cell state estimation, which is the basis of diagnostic use of Raman spectroscopy. Although its nondestructive and ready-to-measure nature is favorable for its application, there remains a major barrier for its practical and general use in life science and medicine [52]. As all molecules have the ability to emit Raman scattering light, the shape of the acquired Raman spectrum depends on the characteristics of the optical components used. It is also affected by the density of the cells and the substrate to which the cells are attached [53]. Therefore, Raman scattering measurement is principally less reproducible than other microscopy methods, and device calibration is required for every measurement. In combination with machine learning, large information hidden in the complicated Raman spectra and Raman images can be utilized. Because machine learning, particularly deep learning, requires a large amount of training data, it is necessary to perform reproducible and high-quality Raman scattering measurements on unknown cells in a high-throughput manner. Furthermore, as with biological and biochemical methods, manual observation is required, which will also cause unfavorable deviations in data quality. We believe that the key to the useful integration of Raman scattering spectroscopy and machine learning is to automate/data-transform the procedures related to microscopic observations, including the establishment of correction methods for modifications due to optical properties and the pretreatment of samples. To overcome these issues, Raman scattering spectroscopy is an alternative to the current method of pathology diagnosis. Recently, several groups reported the prediction of gene expression using Raman spectra in mammalian cells [54,55], which can be further expanded in the future. By prediction of gene expression, it may be possible to determine the type of disease, such as cancer, and select effective medication to treat it by measuring the Raman spectrum and predicting gene expression profiles of the tissue in situ in a clinical environment.

## Figures and Tables

**Figure 1 genes-13-02127-f001:**
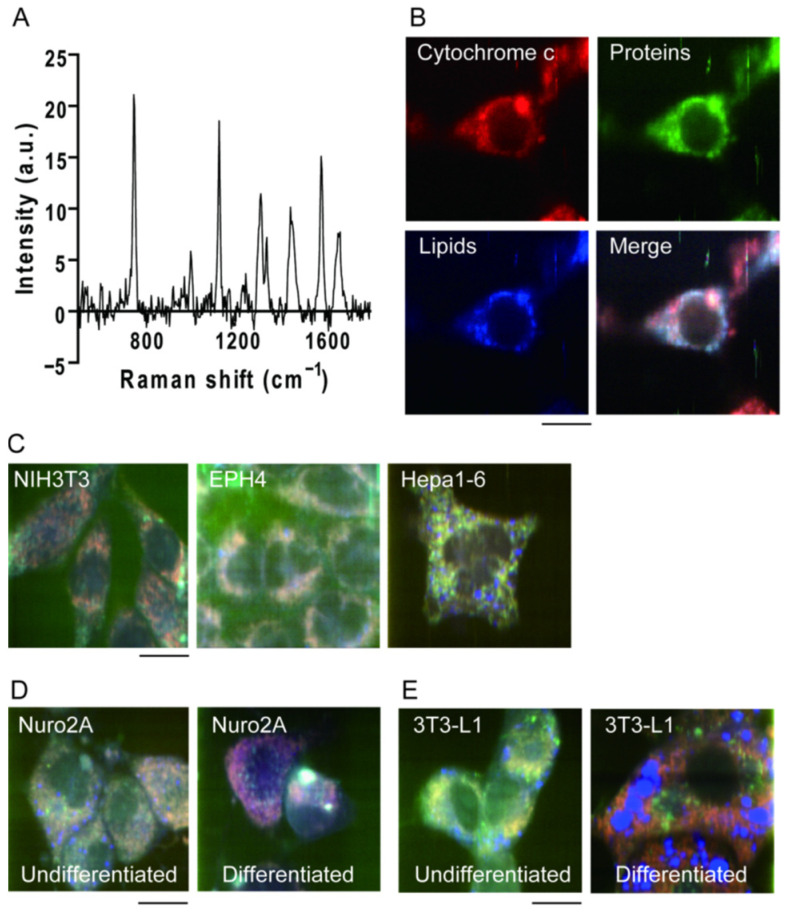
Raman spectra from living cells. (**A**) Typical Raman spectrum from a living NIH3T3 cell. (**B**) Pseudo-color Raman image of NIH3T3 cells. Cells were colored by peak values at 753 cm^−1^ (cytochrome c; red), 1686 cm^−1^ (proteins, green) and 1340 cm^−1^ (lipids; blue). Scale bar, 10 μm. (**C**) Raman image of NIH3T3 cells (left), EPH4 cells (middle), and Hepa1-6 cells (right). Scale bar, 10 μm. (**D**,**E**) Raman image of Neuro 2A (**D**) and 3T3-L1 (**E**) cells before (left) and after (right) differentiation. Scale bar, 10 μm. Reproduction from reference [9].

**Figure 2 genes-13-02127-f002:**
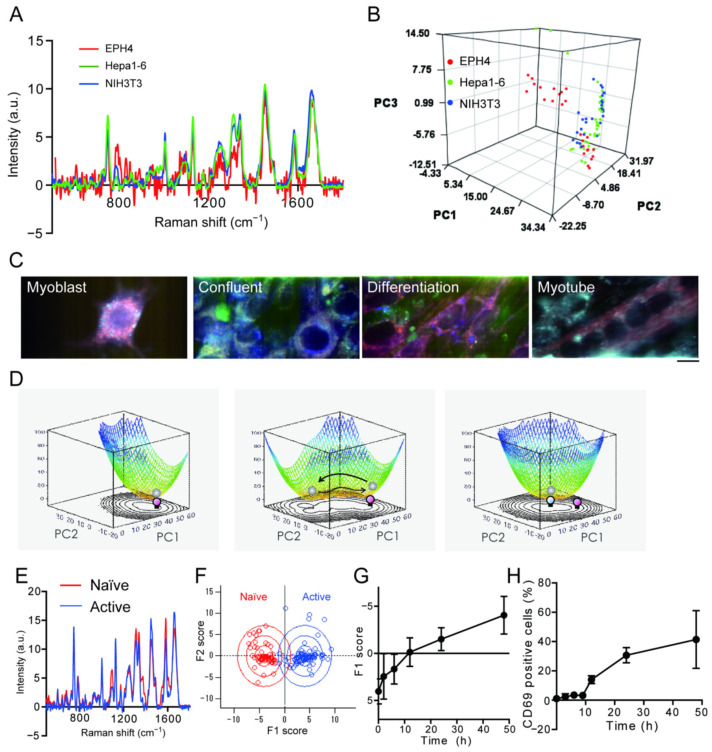
Detection of cell-state change during differentiation using Raman spectra. (**A**) Averaged Raman spectra of EPH4 cells (red), Hepa1-6 cells (green) and NIH3T3 cells (blue). (**B**) Principal component analysis score plot of Raman spectra analysis of 3 cell-lines. (**C**) Raman image of C2C12 cells during differentiation. Scale bar, 10 μm. (**D**) Change in the potential landscapes during C2C12 differentiation estimated by the inverse of logarithm of cell population. The counter plots corresponding to the potential landscapes are also shown. (**E**) Averaged Raman spectra of naïve (red) and active (blue) T cells. (**F**) Results of discriminant analysis of the PCA scores. (**G**) Time course change of F1 score during T cell activation. (**H**) Activation time course of T cell activation assessed by CD69. Reproduction from reference [28].

**Figure 3 genes-13-02127-f003:**
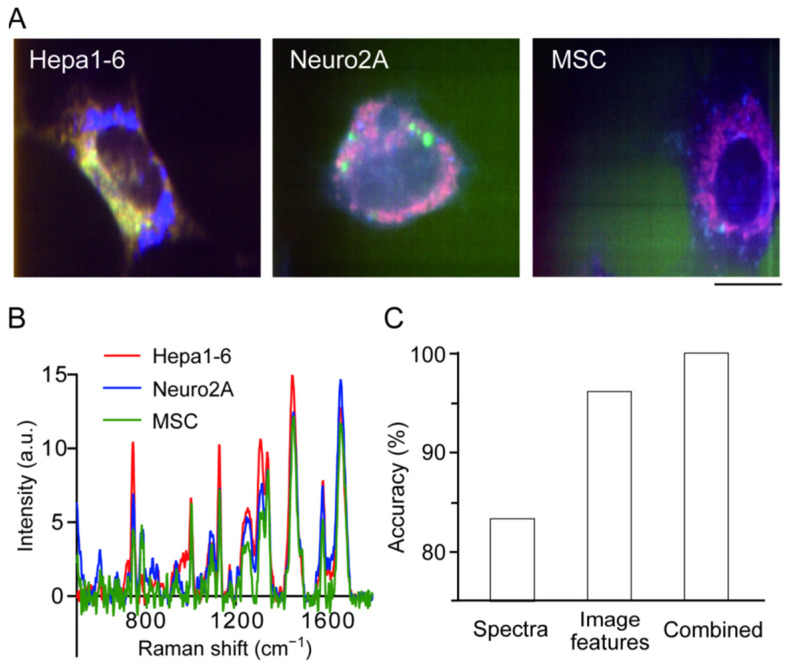
Cell type discrimination using image feature of Raman image. (**A**) Raman image of Hepa1-6 cells (left), Neuro2A cells (middle) and mesenchymal stem cells (MSC) (right). Scale bar, 10 μm. (**B**) Averaged Raman spectra of Hepa1-6 cells (red), Neuro2A cells (blue) and MSC (green). (**C**) Accuracy of cell type discrimination using Raman spectra (left), image features (middle) and both (right). Reproduction from reference [47].

**Figure 4 genes-13-02127-f004:**
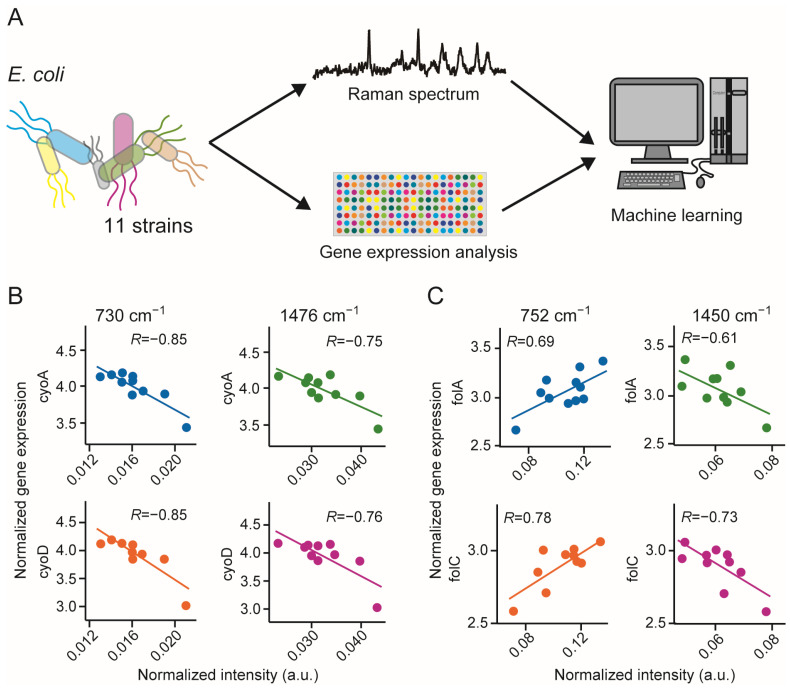
Gene expression prediction using Raman spectra. (**A**) Schematic illustration of the concept of gene expression prediction. (**B**) Relationship between Raman peak intensities at 730 cm^−1^ (left) or 1476 cm^−1^ and guantile-normallized gene expression of *cyoA* (top) or *cyoD* (bottom). (**C**) Relationship between Raman peak intensities at 752 cm^−1^ (left) or 1450 cm^−1^ and gene expression of *folA* (top) or *folC* (bottom). Reproduction from reference [51].

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
