# Peer review of "Recent Advances in Raman Spectral Imaging in Cell Diagnosis and Gene Expression Prediction"

_genes, 2022, doi:10.3390/genes13112127_

Round 1
Reviewer 1 Report
In this manuscript, Watanabe et al., reviewed the basics of Raman spectroscopy and Raman imaging in live cells and cell type discrimination. Also, they reviewed recent attempt to estimate the gene expression profiles from the Raman spectrum of living cells using simple machine learning. Although this review is short and might not cover a lot of papers in the related field, this reviewer thinks that it would be a good review for beginners to know the basics of the Raman spectral imaging in cell diagnosis and gene expression prediction.
#1
This manuscript explained the basics of Raman spectral imaging by using the papers from the authors’ group. This reviewer recommends the authors to introduce some other published review papers covering more papers in the beginning of the manuscript.
#2
In the section 1-4, papers using mammalian cells were cited to explain each topic. However, paper using E.coli was used in the section 5. Thus, there is a gap between them. If possible, please fill the gap between them by explaining the technical difference and similarities for both cases with citing the appropriate papers.
#3
Original source of Figures 1-4 should be described in the figure legends.
#4
“E. Coli.” In Fig. 4A should be “E. coli”
#5
Scale bar is missing for Figs. 1C-E and 2C.
Author Response
We appreciate the reviewer’s helpful comments, and have revised our manuscript accordingly. We modified our manuscript after careful consideration of the reviewer’s comments and improved the points raised by the reviewer. Our point-by-point response to each comment is as follows.
#1: This manuscript explained the basics of Raman spectral imaging by using the papers from the authors’ group. This reviewer recommends the authors to introduce some other published review papers covering more papers in the beginning of the manuscript.
We have largely revised the reference and review papers covering more papers are added. Thank you very much for your suggestion.
#2: In the section 1-4, papers using mammalian cells were cited to explain each topic. However, paper using E.coli was used in the section 5. Thus, there is a gap between them. If possible, please fill the gap between them by explaining the technical difference and similarities for both cases with citing the appropriate papers.
We have added sentence to justify the use of E. coli in gene expression estimation using Raman spectra in the revised manuscript.
#3: Original source of Figures 1-4 should be described in the figure legends.
We have added the original sources of Figs. 1-4 in the figure legends in the revised manuscript.
#4: “E. Coli.” In Fig. 4A should be “E. coli”
“E. Coli.” In Fig. 4A is now corrected to “E. coli” in the revised manuscript.
#5: Scale bar is missing for Figs. 1C-E and 2C.
Scale bars are now added in Figs. 1C-E and 2C.
Reviewer 2 Report
The manuscript describes the comprehensive works by the authors on the subcellular changes detected by Raman Spectroscopy. There are several concerns about the manuscript, as follows:
1) Overall, the manuscript reported results and discussions from the combinations of several original research conducted by the authors themselves, as the authors also used the terms "We attempted, We measured, We show...". Thus, the type of the manuscript should be changed to Original Article, with the appropriate formats and Methods. For examples:
a) What are the protocols utilized to generate cell images with different colour tone using Raman spectroscopy?
b) How do you measure Accuracy % in Figure 3?
2) The references used were not enough to categorized the manuscript as a Review paper. Even for the Original Article, more reference should be added in the Discussions of the results as the works by the authors are not the first research on the cell imaging by Raman.
a) For example, in page 7, "mammalian cell has more than 20,000 genes, but because cell functions are a result of the complex interaction of these genes, the cell functions can be aggregated and classified into only ~300 different types.", what are the source for this data?
b) Out of 16, only 5 cited references are the recent publications (within the last 5 years). Authors should make more comparisons of the works in the manuscript with other previous research, and emphasize the novelty of the present data.
3) Page 4, at last 2 sentences, "The result shows that both the differences in spectra depending on the cell type..." What do you mean by "both"? as the paragraph described the results of 3 cell lines.
4) Figure 4, what is the label for the X-axis of the graphs?
5) The conclusions drawn in subsection 6 only support the findings in subsection 5. Authors should conclude the overall works described in the manuscript.
Author Response
We appreciate the reviewer’s helpful comments, and have revised our manuscript accordingly. We modified our manuscript after careful consideration of the reviewer’s comments and improved the points raised by the reviewer. Our point-by-point response to each comment is as follows.
1) Overall, the manuscript reported results and discussions from the combinations of several original research conducted by the authors themselves, as the authors also used the terms "We attempted, We measured, We show...". Thus, the type of the manuscript should be changed to Original Article, with the appropriate formats and Methods. For examples:
a) What are the protocols utilized to generate cell images with different colour tone using Raman spectroscopy?
b) How do you measure Accuracy % in Figure 3?
This is a review article and we should not use “We” in this manuscript. We have revised the manuscript and first-person sentences are now removed.
2) The references used were not enough to categorized the manuscript as a Review paper. Even for the Original Article, more reference should be added in the Discussions of the results as the works by the authors are not the first research on the cell imaging by Raman.
a) For example, in page 7, "mammalian cell has more than 20,000 genes, but because cell functions are a result of the complex interaction of these genes, the cell functions can be aggregated and classified into only ~300 different types.", what are the source for this data?
b) Out of 16, only 5 cited references are the recent publications (within the last 5 years). Authors should make more comparisons of the works in the manuscript with other previous research, and emphasize the novelty of the present data.
We have revised the references and source for the data presented are now included. We also added recent publications where 12 out of 39 papers are within last 5 years.
3) Page 4, at last 2 sentences, "The result shows that both the differences in spectra depending on the cell type..." What do you mean by "both"? as the paragraph described the results of 3 cell lines.
The word “both” indicate ‘the cell type’ and ‘heterogeneity’. Because this word “both” may confuse the readers, we have deleted this word in the revised manuscript.
4) Figure 4, what is the label for the X-axis of the graphs?
X-axis in Fig. 4 is quantile-normalized gene expression. This is now stated in the figure legend of the revised manuscript.
5) The conclusions drawn in subsection 6 only support the findings in subsection 5. Authors should conclude the overall works described in the manuscript.
We have largely revised subsection 6 to cover overall works described in this manuscript.
Round 2
Reviewer 2 Report
The comments from the previous report are revised. However, there are a point at which the author should carefully revised.
As I commented previously, authors should make more comparisons of the works with other research, and identify the differences or similarities. It seems that the manuscript reviews and discusses the past works that had been conducted by the authors since 2012 (reference of number 9, 17, 19, 25, 26, 33, and 35). They are relatable and appropriate to the title and content, but what are the differences or similarities with the other research?
Most of references from other researchers are cited for the sentences related to theories, not the works/results comparisons.
In section 2 and 4, authors can review more works that other research conducted on other types of cells or models.
In section 3, authors can review more works that other research conducted on other types of machine-learning.
This review manuscript should offer a comprehensive analysis of the existing literature, within the 10 years time, as term "Recent" used in the title. Using Raman spectroscopy for cell diagnosis is not new in this field.
Author Response
In section 2 and 4, authors can review more works that other research conducted on other types of cells or models.
We reviewed more works relating to other cell types or models in the revised manuscript.
In section 3, authors can review more works that other research conducted on other types of machine-learning.
In the revised manuscript, we mentioned other machine-learning methods used in Raman spectral analysis in section 3.
This review manuscript should offer a comprehensive analysis of the existing literature, within the 10 years time, as term "Recent" used in the title. Using Raman spectroscopy for cell diagnosis is not new in this field.
According to the reviewer’s comment, we have added recent references within 10 years’ time in the revised manuscript (36 references within 10 years out of 55).